# SARS-CoV-2 shedding dynamics across the respiratory tract, sex, and disease severity for adult and pediatric COVID-19

Paul Z Chen[1], Niklas Bobrovitz[2,3,4], Zahra A Premji[5], Marion Koopmans[6], David N Fisman[7], Frank X Gu[1,8]*

[1]Department of Chemical Engineering & Applied Chemistry, University of Toronto, Toronto, Canada; [2]Temerty Faculty of Medicine, University of Toronto, Toronto, Canada; [3]Department of Critical Care Medicine, Cumming School of Medicine, University of Calgary, Calgary, Canada; [4]O'Brien Institute of Public Health, University of Calgary, Calgary, Canada; [5]Libraries, University of Victoria, Victoria, Canada; [6]Department of Viroscience, Erasmus University Medical Center, Rotterdam, Netherlands; [7]Division of Epidemiology, Dalla Lana School of Public Health, University of Toronto, Toronto, Canada; [8]Institute of Biomedical Engineering, University of Toronto, Toronto, Canada

## Abstract

**Background:** Previously, we conducted a systematic review and analyzed the respiratory kinetics of severe acute respiratory syndrome coronavirus 2 (SARS-CoV-2) (Chen et al., 2021). How age, sex, and coronavirus disease 2019 (COVID-19) severity interplay to influence the shedding dynamics of SARS-CoV-2, however, remains poorly understood.

**Methods:** We updated our systematic dataset, collected individual case characteristics, and conducted stratified analyses of SARS-CoV-2 shedding dynamics in the upper (URT) and lower respiratory tract (LRT) across COVID-19 severity, sex, and age groups (aged 0–17 years, 18–59 years, and 60 years or older).

**Results:** The systematic dataset included 1266 adults and 136 children with COVID-19. Our analyses indicated that high, persistent LRT shedding of SARS-CoV-2 characterized severe COVID-19 in adults. Severe cases tended to show slightly higher URT shedding post-symptom onset, but similar rates of viral clearance, when compared to nonsevere infections. After stratifying for disease severity, sex and age (including child vs. adult) were not predictive of respiratory shedding. The estimated accuracy for using LRT shedding as a prognostic indicator for COVID-19 severity was up to 81%, whereas it was up to 65 % for URT shedding.

**Conclusions:** Virological factors, especially in the LRT, facilitate the pathogenesis of severe COVID-19. Disease severity, rather than sex or age, predicts SARS-CoV-2 kinetics. LRT viral load may prognosticate COVID-19 severity in patients before the timing of deterioration and should do so more accurately than URT viral load.

**Funding:** Natural Sciences and Engineering Research Council of Canada (NSERC) Discovery Grant, NSERC Senior Industrial Research Chair, and the Toronto COVID-19 Action Fund.

*For correspondence: f.gu@utoronto.ca

## Introduction

As of August 8, 2021, the coronavirus disease 2019 (COVID-19) pandemic has caused more than 202.6 million infections and 4.2 million deaths globally (*Dong et al., 2020*). The clinical spectrum of

COVID-19, caused by severe acute respiratory syndrome coronavirus 2 (SARS-CoV-2), is wide, ranging from asymptomatic infection to fatal disease. For cases that deteriorate into severe COVID-19, deterioration occurs, on median, 10 days after symptom onset (*Solomon et al., 2020*; *Zhou et al., 2020*). Risk factors for severe illness and death include age, sex, smoking, and comorbidities, such as obesity, hypertension, diabetes, and cardiovascular disease (*Onder et al., 2020*; *Tartof et al., 2020*; *Zhou et al., 2020*). Emerging evidence indicates that age and sex differences in innate, cross-reactive, and adaptive immunity facilitate the higher risks observed in older and male cases (*Rydyznski Moderbacher et al., 2020*; *Ng et al., 2020*; *Pierce et al., 2020*; *Takahashi et al., 2020*). Conversely, robust immune responses putatively mediate nonsevere illness, in part, by controlling the replication of SARS-CoV-2 (*Lucas et al., 2021*; *Lucas et al., 2020*).

As a respiratory virus, the shedding dynamics of SARS-CoV-2 in the upper (URT) and lower respiratory tract (LRT) provide insight into the clinical and epidemiological characteristics of COVID-19. URT viral load has been associated with transmission risk, duration of infectiousness, disease severity, and mortality (*Chen et al., 2021b*; *Fu et al., 2021*; *Magleby et al., 2020*; *Marks et al., 2021*; *Pujadas et al., 2020*; *van Kampen et al., 2021*; *Westblade et al., 2020*; *Wölfel et al., 2020*). Key questions, however, remain. While chest computed tomography (CT) evidence of viral pneumonitis suggests pulmonary replication in most symptomatic cases (*Bernheim et al., 2020*), the LRT kinetics of SARS-CoV-2, especially as related to disease severity, remain unclear. How age, sex, and disease severity influence shedding dynamics is poorly understood, especially for children. Moreover, it is unclear whether respiratory viral load can accurately predict COVID-19 severity, with conflicting results from analyses of low sample numbers (*Argyropoulos et al., 2020*; *Lucas et al., 2020*; *Pujadas et al., 2020*; *Silva et al., 2021*; *Walsh et al., 2020*; *Westblade et al., 2020*).

For insight into these questions, we conducted a systematic review on SARS-CoV-2 quantitation from respiratory specimens and developed a large, diverse dataset of viral loads and individual case characteristics. Stratified analyses then assessed SARS-CoV-2 shedding dynamics across the respiratory tract, age, sex, and COVID-19 severity.

## Materials and methods
### Data sources and searches
Our systematic review identified studies reporting SARS-CoV-2 quantitation in respiratory specimens taken during the estimated infectious period (−3 to 10 days from symptom onset [DFSO]) (*He et al., 2020*; *Wölfel et al., 2020*). The systematic review protocol was based on our previous study (*Chen et al., 2021a*) and was prospectively registered on PROSPERO (registration number, CRD42020204637). The systematic review was conducted according to Cochrane methods guidance (*Higgins et al., 2019*). PRISMA reporting guidelines were followed (*Moher et al., 2009*).

Up to November 20, 2020, we searched, without the use of filters or language restrictions, the following sources: MEDLINE (Ovid, 1946 to November 20, 2020, *Wyllie et al., 2020*), EMBASE (Ovid, 1974 to November 20, 2020, *Wyllie et al., 2020*), Cochrane Central Register of Controlled Trials (via Ovid, 1991 to November 20, 2020, *Wyllie et al., 2020*), Web of Science Core Collection (up to November 20, 2020, *Wyllie et al., 2020*), and medRxiv and bioRxiv (both searched through Google Scholar via the Publish or Perish program, up to November 20, 2020, *Wyllie et al., 2020*). We also gathered studies by searching through the reference lists of review articles identified by the database search, by searching through the reference lists of included articles, through expert recommendation (by Eric J Topol and Akiko Iwasaki on Twitter) and by hand-searching through journals. A comprehensive search was developed by a librarian (ZP). The line-by-line search strategies for all databases are included in *Figure 1—source data 1* to 5. The search results were exported from each database and uploaded to the Covidence online system (research resource identifier, RRID:SCR_016484) for deduplication and screening.

### Study selection
Studies that reported SARS-CoV-2 quantitation in individual URT (nasopharyngeal swab [NPS], nasopharyngeal aspirate, oropharyngeal swab [OPS], or posterior oropharyngeal saliva [POS]) or LRT (endotracheal aspirate [ETA] or sputum [Spu]) specimens taken during the estimated infectious period (−3 to 10 DFSO) in humans were included (additional details given in the Appendix). As semiquantitative metrics (cycle threshold [Ct] values) cannot be compared on an absolute scale between studies

based on instrument and batch variation (*Han et al., 2021*), studies reporting specimen measurements as Ct values, without quantitative calibration, were excluded. Two authors (PZC and NB) independently screened titles and abstracts and reviewed full texts. At the full-text stage, reference lists were reviewed for study inclusion. Inconsistencies were resolved by discussion and consensus, and 26 studies met the inclusion criteria (*Bal et al., 2020*; *Benotmane et al., 2020*; *Biguenet et al., 2021*; *Fajnzylber et al., 2020*; *Han et al., 2020*; *Hirotsu et al., 2020*; *Hurst et al., 2020*; *Iwasaki et al., 2020*; *L'Huillier et al., 2020*; *Lavezzo et al., 2020*; *Pan et al., 2020*; *Peng et al., 2020*; *Shrestha et al., 2020*; *Sun et al., 2020*; *To et al., 2020*; *van Kampen et al., 2021*; *Vetter et al., 2020*; *Wölfel et al., 2020*; *Wyllie et al., 2020*; *Xu et al., 2020a*; *Yazdanpanah, 2021*; *Yilmaz et al., 2021*; *Yonker et al., 2020*; *Zhang et al., 2021*; *Zheng et al., 2020*; *Zou et al., 2020*). Additional details on study selection can be found in our previous protocol (*Chen et al., 2021a*).

## Data extraction and risk-of-bias assessment

Two authors (PZC and NB) independently collected data (specimen measurements taken between –3 and 10 DFSO, specimen type, volume of viral transport media [VTM], and case characteristics, including age, sex, and disease severity) from contributing studies and assessed risk of bias using a modified version of the Joanna Briggs Institute (JBI) tools for case series, analytical cross-sectional studies, and prevalence studies (*Moola et al., 2020*; *Munn et al., 2019*; *Munn et al., 2015*) (shown in the Appendix). Items were judged with responses to data inquiries, if authors responded.

Data were collected for individually reported specimens of known type, with known DFSO, and for COVID-19 cases with known age, sex or severity. Case characteristics were collected directly from contributing studies when reported individually or obtained via data request from the authors. Data from serially sampled asymptomatic cases were included, and the day of laboratory diagnosis was referenced as 0 DFSO (*Lavezzo et al., 2020*; *Wölfel et al., 2020*). Based on the modified JBI checklist, studies were considered to have low risk of bias if they met the majority of items and included item 1 (representative sample). Discrepancies were resolved by discussion and consensus. Studies at high or unclear risk of bias typically included samples that were not representative of the target population; did not report the VTM volume used; had non-consecutive inclusion for case series and cohort studies or did not use probability-based sampling for cross-sectional studies; and did not report the response rate.

## Respiratory viral load

To enable analyses based on respiratory viral load (rVL, viral RNA concentration in the respiratory tract) and to account for between-study variation in specimen measurements, the rVL for each collected sample was estimated based on the specimen concentration (viral RNA concentration in the specimen) and its dilution factor in VTM. Typically, swabbed specimens (NPS and OPS) report the viral RNA concentration in VTM. Based on the VTM volume reported in the study along with the expected uptake volume for swabs (0.128 ± 0.031 ml, mean ± SD) (*Warnke et al., 2014*), we calculated the dilution factor for each respiratory specimen and then estimated the rVL. Similarly, liquid specimens (ETA, POS, and Spu) are often diluted in VTM, and the rVL was estimated based on the reported collection and VTM volumes. If the diluent volume was not reported, then VTM volumes of 1 ml (NPS and OPS) or 2 ml (POS and ETA) were assumed (*Lavezzo et al., 2020*; *To et al., 2020*). Unless dilution was reported, Spu specimens were taken as undiluted (*Wölfel et al., 2020*). The non-reporting of VTM volume was noted as an element increasing risk of bias in the modified JBI critical appraisal checklist. For laboratory-confirmed COVID-19 cases, negative specimen measurements were taken at the reported assay detection limit in the respective study.

## Case definitions

As severity in the clinical manifestations of COVID-19 and case-fatality rates tend to increase among children (aged 0–17 years), younger adults (aged 18–59 years), and older adults (aged 60 years or older) (*Onder et al., 2020*; *Zheng et al., 2020*), the data were delineated by these three age groups. Cases were also categorized by sex.

U.S. National Institutes of Health guidance was used to categorize disease severity as nonsevere or severe (*National Institutes of Health, 2021*) (*Appendix 1—table 1*). The nonsevere group included those with asymptomatic infection (individuals who test positive via a molecular test for SARS-CoV-2 and report no symptoms consistent with COVID-19); mild illness (individuals who report any signs or

symptoms of COVID-19, including fever, cough, sore throat, malaise, headache, muscle pain, nausea, vomiting, diarrhea, loss of taste and smell, but who do not have dyspnea or abnormal chest imaging); and moderate illness (individuals with clinical or radiographic evidence of LRT disease, fever >39.4 °C or $SpO_2$ >94% on room air) disease. The severe group included those with severe illness (individuals who have $SpO_2$ <94% on room air, $[PaO_2/FiO_2]$ < 300 mmHg, respiratory rate >30 breaths/min or lung infiltrates > 50%) and critical illness (respiratory failure, septic shock, or multiple organ dysfunction).

## Regression analyses

To assess the respiratory shedding of SARS-CoV-2 and compare age, sex, or severity groups, we analyzed the data via normal linear regression (*Hurst et al., 2020*; *Lucas et al., 2020*). Previous studies have shown that SARS-CoV-2 shedding tends to diminish exponentially after 1 DFSO in the URT and, at least, after 4 DFSO in the LRT (*Bernheim et al., 2020*; *Chen et al., 2021a*; *Wölfel et al., 2020*). Although LRT shedding may peak before 4 DFSO, there is limited data near or before symptom onset. Hence, rVLs (in units of $\log_{10}$ copies/ml) between 1 and 10 DFSO for the URT, or 4 and 10 DFSO for the LRT, were fitted using linear regression with interaction:

$$V = \alpha + \beta_1 X_1 + \beta_2 X_2 + \beta_3 X_1 X_2, \qquad (1)$$

where $V$ represents the rVL, $\alpha$ represents the estimated mean rVL (at 1 DFSO for URT or 4 DFSO for LRT) for the reference group, $X_1$ represents DFSO for the reference group, $X_2$ represents the comparison group, $\beta_1$ represents the effect of DFSO on rVL for the reference group, $\beta_2$ represents the effect of the comparison group on the main effect (mean rVL at 1 DFSO), and $\beta_3$ represents the interaction between DFSO and groups. Regression analyses were offset by DFSO such that mean rVLs at 1 DFSO for URT, or 4 DFSO for LRT, were compared between groups by the main effect (i.e., effect on the intercept in the regression $t$-test for $\beta_2$). Shedding dynamics were compared between groups by interaction (regression $t$-test for $\beta_3$). The statistical significance of viral clearance for each group was analyzed using simple linear regression (regression $t$-test on the slope). Each group in statistical analyses included all rVLs for which the relevant characteristic (LRT or URT, age group, sex, or disease severity) was ascertained at the individual level. Groups with small sample sizes were not compared, as these analyses are more sensitive to potential sampling error.

Regression models were extrapolated to 0 $\log_{10}$ copies/ml to estimate the total duration of shedding. Some clinical studies report shedding duration based on assay negativity, when the viral RNA concentration in the specimen reaches the detection limit of the assay (often between 1 and 4 $\log_{10}$ copies/ml), and these cases may continue to shed viral RNA. To show the relationship between the two approaches, we used our regression model for URT shedding and estimated the shedding duration to a specimen concentration of 3 $\log_{10}$ copies/ml when sampling was conducted with nasopharyngeal swabs (approximately equivalent to an rVL of 2.1 $\log_{10}$ copies/ml). Then, the estimated mean duration of URT shedding for severe cases was 20.8 (95% CI: 14.5–27.0) DFSO, while it was 20.3 (95% CI: 16.8–23.7) DFSO for nonsevere cases. These values are in line with those reported by studies considering the assay detection limit (*Cevik et al., 2021*), supporting our regression models, and can be compared with those reported in the body text.

Statistical analyses were performed using OriginPro 2019b (RRID:SCR_014212, OriginLab) and the General Linear regression app. p-Values below 0.05 were considered statistically significant.

## Distribution analyses

Previously, our analyses found that SARS-CoV-2 rVLs best conform to Weibull distributions (*Chen et al., 2021b*). To assess heterogeneity in shedding in this study, rVL data were fitted to Weibull distributions. The Weibull quantile function and Weibull cumulative distribution function were used to estimate the rVL at a case percentile and the percentage of cases at a given rVL, respectively. Each distribution was fitted to groups that included all rVLs for which the relevant characteristic (LRT or URT, age group, sex, or disease severity) was ascertained at the individual level. Distribution fitting was performed using Matlab R2019b (RRID:SCR_001622, MathWorks) and the Distribution Fitter app.

## Prognostication accuracy

The fitted Weibull distributions were used to estimate the accuracy when using URT or LRT rVLs of SARS-CoV-2 as a prognostic indicator for COVID-19 severity. The overlapped area under the curve

(AUC) and separated AUC were calculated using the rVL distributions for severe and nonsevere adult COVID-19. These calculations were performed for each DFSO and, separately, for the URT and LRT. The estimated maximal accuracy for prognostication at a given rVL threshold was then estimated by $A = 50 + \left(50 * AUC_{separated}\right)$, where $AUC_{separated}$ represents the AUC that was separated for the nonsevere and severe distributions. The 95 % CIs for prognostication accuracy were estimated using the proportional 95 % CIs in the respective Weibull cumulative distributions. As the Weibull cumulative distributions estimate the percentage of cases at a given rVL, they were also used to estimate the sensitivity and specificity at a given prognostic threshold of rVL. The cases with rVL lower than the prognostic threshold were predicted to have nonsevere COVID-19, whereas those with rVL above it were predicted to have severe COVID-19. Hence, we used the cumulative distributions for nonsevere and severe adult cases on a DFSO and calculated the proportion of cases that were true positive, false positive, false negative, and true negative rates across prognostic thresholds of rVL. Sensitivity and specificity were calculated based on these values. These analyses were coded in Matlab R2019b (RRID:SCR_001622, MathWorks) and are available at GitHub (copy archived at swh:1:rev:c96390f98f47f17939f3669c7c8fad96f9603e84, *Chen, 2019*).

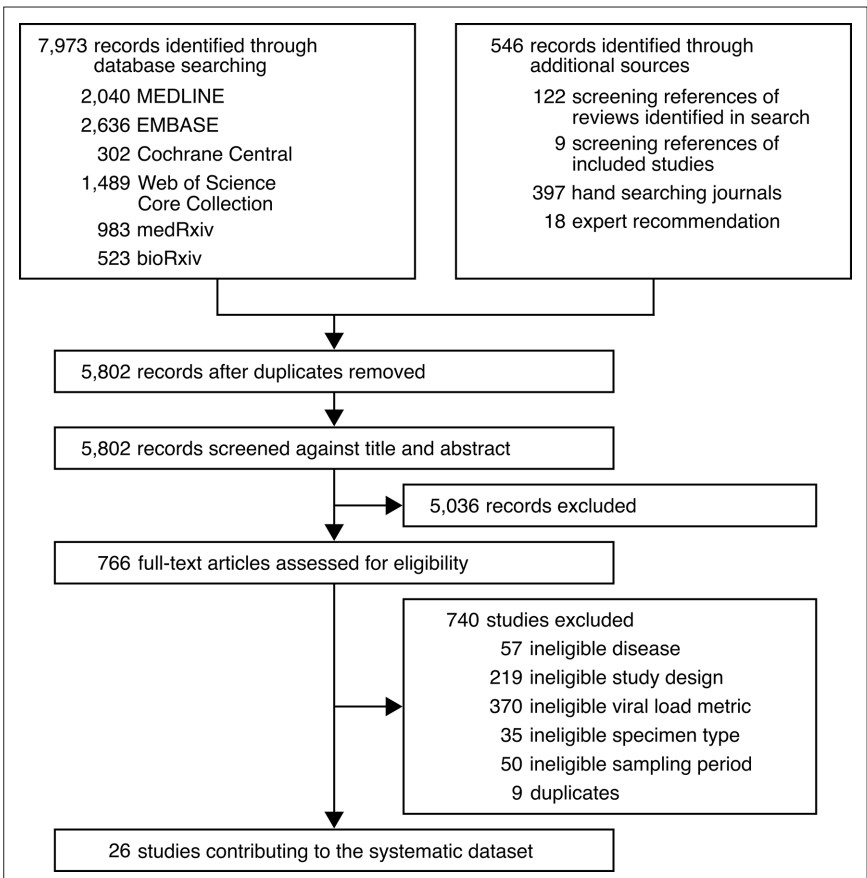

**Figure 1.** Study selection.

The online version of this article includes the following figure supplement(s) for figure 1:

**Source data 1.** Search strategy used for MEDLINE.

**Source data 2.** Search strategy used for EMBASE.

**Source data 3.** Search strategy used for Cochrane Central.

**Source data 4.** Search strategy used for Web of Science Core Collection.

**Source data 5.** Search strategy used for medRxiv and bioRxiv.

**Figure supplement 1.** Summary of respiratory viral loads in the systematic dataset.

**Table 1.** Characteristics of adult and pediatric coronavirus disease 2019 (COVID-19) cases in the systematic dataset.

| | Adult | Pediatric |
|---|---|---|
| Cases, n | 1266 | 136 |
| URT specimens, n | 1513 | 192 |
| LRT specimens, n | 210 | 0 |
| Mean age (SD), years | 51.8 (18.0) | 8.7 (5.3) |
| Male, n (%) | 528 (44.0) | 63 (52.5) |
| **Disease severity, n (%)** | | |
| Asymptomatic | 2 (0.2) | 5 (3.7) |
| Mild | 710 (57.5) | 112 (83.6) |
| Moderate | 178 (14.4) | 17 (12.7) |
| Severe | 167 (13.5) | 0 (0.0) |
| Critical | 178 (14.4) | 0 (0.0) |

Adult cases were those aged 18 years or older, while pediatric cases were those aged younger than 18 years. Upper respiratory tract = URT. Lower respiratory tract = LRT.

# Results

## Overview of contributing studies

The systematic search (*Figure 1—source data 1*, *Figure 1—source data 2*, *Figure 1—source data 3*, *Figure 1—source data 4*, *Figure 1—source data 5*) identified 5802 deduplicated results. After screening and full-text review, 26 studies met the inclusion criteria, and data were collected for individually reported specimens of known type and taken on a known DFSO for COVID-19 cases with known age, sex, or severity (*Figure 1*). From 1402 COVID-19 cases, we collected 1915 quantitative specimen measurements (viral RNA concentration in a respiratory specimen) of SARS-CoV-2 (*Table 1*) and used them to estimate rVLs (viral RNA concentration in the respiratory tract) (*Figure 1—figure supplement 1*). For pediatric cases, the search found only nonsevere infections and URT specimen measurements. *Appendix 1—table 1* and *Appendix 1—table 2* summarize the characteristics of contributing studies, of which 18 had low risk of bias according to the modified JBI critical appraisal checklist.

## URT shedding of SARS-CoV-2 for adult COVID-19

To interpret the complex interplay between SARS-CoV-2 shedding dynamics and age, sex, and COVID-19 severity, we stratified our systematic dataset into age, sex, and severity groups and then conducted a series of linear regression analyses. For adult COVID-19, regression analysis showed that the mean URT rVL at 1 DFSO was significantly greater (p = 0.005) for severely infected cases (8.28 [95% CI: 7.71–8.84] $\log_{10}$ copies/ml) than nonsevere ones (7.45 [95% CI: 7.26–7.65] $\log_{10}$ copies/ml) (*Figure 2A, D,*). Meanwhile, these groups showed comparable URT dynamics post-symptom onset (p for interaction = 0.479), as severe adult cases tended to cleared SARS-CoV-2 from the URT at –0.31 (95% CI: –0.40 to–0.22) $\log_{10}$ copies/ml/day while nonsevere ones did so at –0.28 (95% CI: –0.32 to –0.24) $\log_{10}$ copies/ml/day (*Figure 2A, E*). For severe cases, the estimated mean duration of URT shedding (down to 0 $\log_{10}$ copies/ml) was 27.5 (95% CI: 21.2–33.8) DFSO; it was 27.9 (95% CI: 24.4–31.3) DFSO for nonsevere cases.

After stratifying adults for disease severity, our analyses showed no significant differences in URT shedding levels or dynamics between sex or age groups (*Figure 2D,E*, *Figure 2—figure supplement 1*). For severe disease, male and female cases had comparable mean rVLs at 1 DFSO (p = 0.326) and rates of SARS-CoV-2 clearance (p for interaction = 0.280). Similarly, for nonsevere illness, male and female cases had no significant difference in mean rVL at 1 DFSO (p = 0.085) or URT dynamics (p for interaction = 0.644). For nonsevere illness, younger and older adults had no significant difference in URT shedding levels at 1 DFSO (p = 0.294) or post-symptom onset dynamics (p for interaction = 0.100). For severe disease, the adult age groups showed similar mean rVLs at 1 DFSO (p = 0.915) and rates of viral clearance (p for interaction = 0.359).

## URT shedding of SARS-CoV-2 for pediatric COVID-19

For pediatric COVID-19, regression estimated that, in the URT, the mean rVL at 1 DFSO was 7.32 (95% CI: 6.78–7.86) $\log_{10}$ copies/ml and the SARS-CoV-2 clearance rate was –0.32 (95% CI: –0.42 to –0.22) $\log_{10}$ copies/ml/day (*Figure 2F,G*). Both estimates were comparable between the sexes for children (*Figure 2—figure supplement 1*). The estimated mean duration of URT shedding (down to 0 $\log_{10}$ copies/ml) was 22.6 (95% CI: 17.0–28.1) DFSO for children with COVID-19.

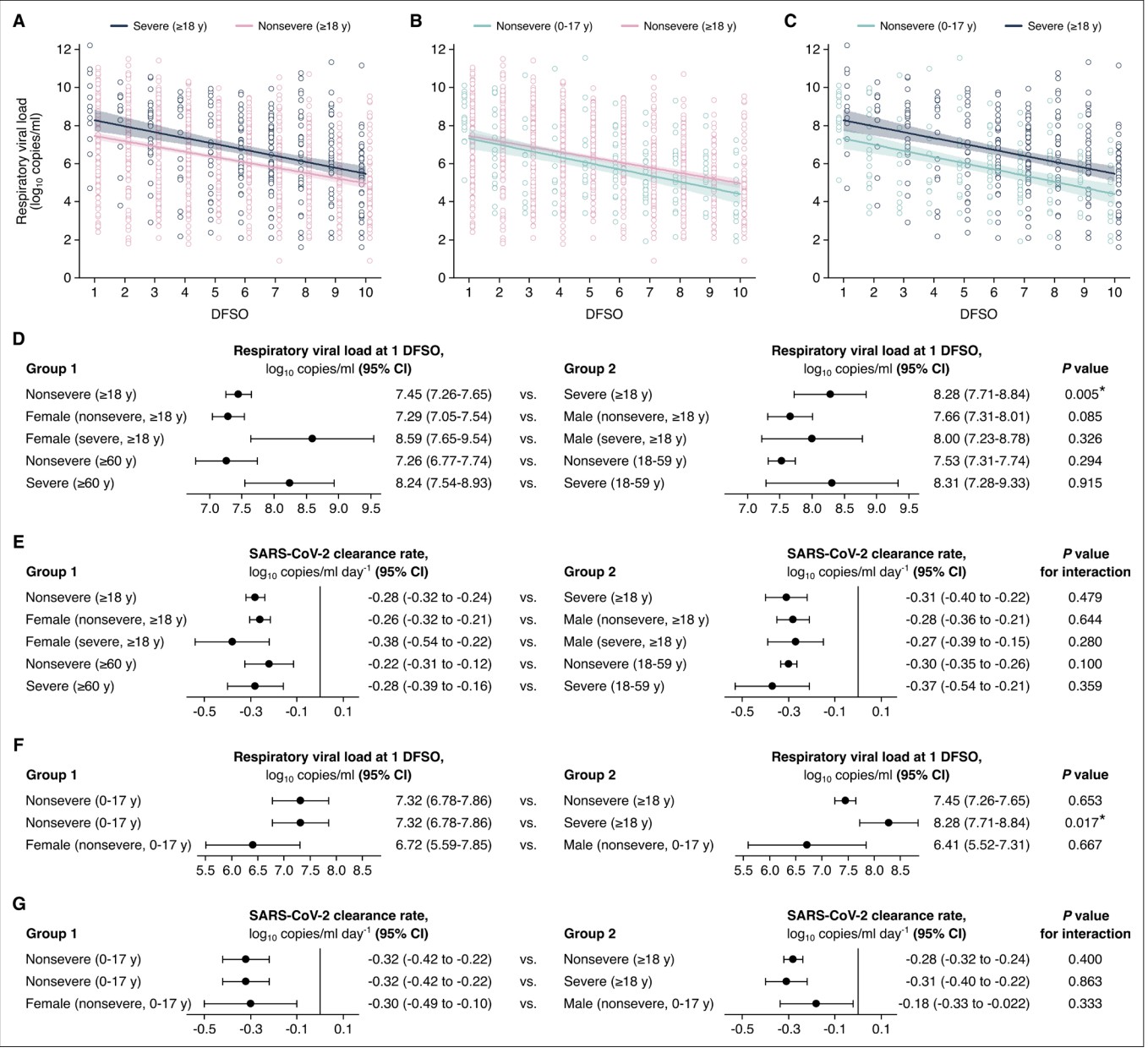

**Figure 2.** Comparison of severe acute respiratory syndrome coronavirus 2 (SARS-CoV-2) shedding in the upper respiratory tract (URT) across severity, sex, and age groups. (**A–C**) URT shedding for severe and nonsevere adult (aged 18 years or older) coronavirus disease 2019 (COVID-19) (**A**), for nonsevere pediatric (aged 0–17 years) and nonsevere adult COVID-19 (**B**) and for nonsevere pediatric and severe adult COVID-19 (**C**). Open circles represent respiratory viral load (rVL) data and were offset from their day from symptom onset (DFSO) for visualization. Lines and bands show regressions and their 95 % CIs, respectively. (**D and E**) Comparisons of URT shedding levels at 1 DFSO (**D**) and URT shedding dynamics (**E**) between severity, age, and sex groups for COVID-19. (**F and G**) Comparisons of URT shedding levels at 1 DFSO (**F**) and URT shedding dynamics (**G**) between pediatric and adult groups for COVID-19. The black line in (**E**) and (**G**) depicts 0, the threshold for no significant trend in SARS-CoV-2 clearance. Linear regression analyses with interaction determined p-values and compared shedding levels and dynamics between the two groups in each row.

The online version of this article includes the following figure supplement(s) for figure 2:

**Figure supplement 1.** Severe acute respiratory syndrome coronavirus 2 (SARS-CoV-2) shedding in the upper respiratory tract (URT) for adult coronavirus disease 2019 (COVID-19).

**Figure supplement 2.** Severe acute respiratory syndrome coronavirus 2 (SARS-CoV-2) shedding in the upper respiratory tract (URT) for pediatric coronavirus disease 2019 (COVID-19).

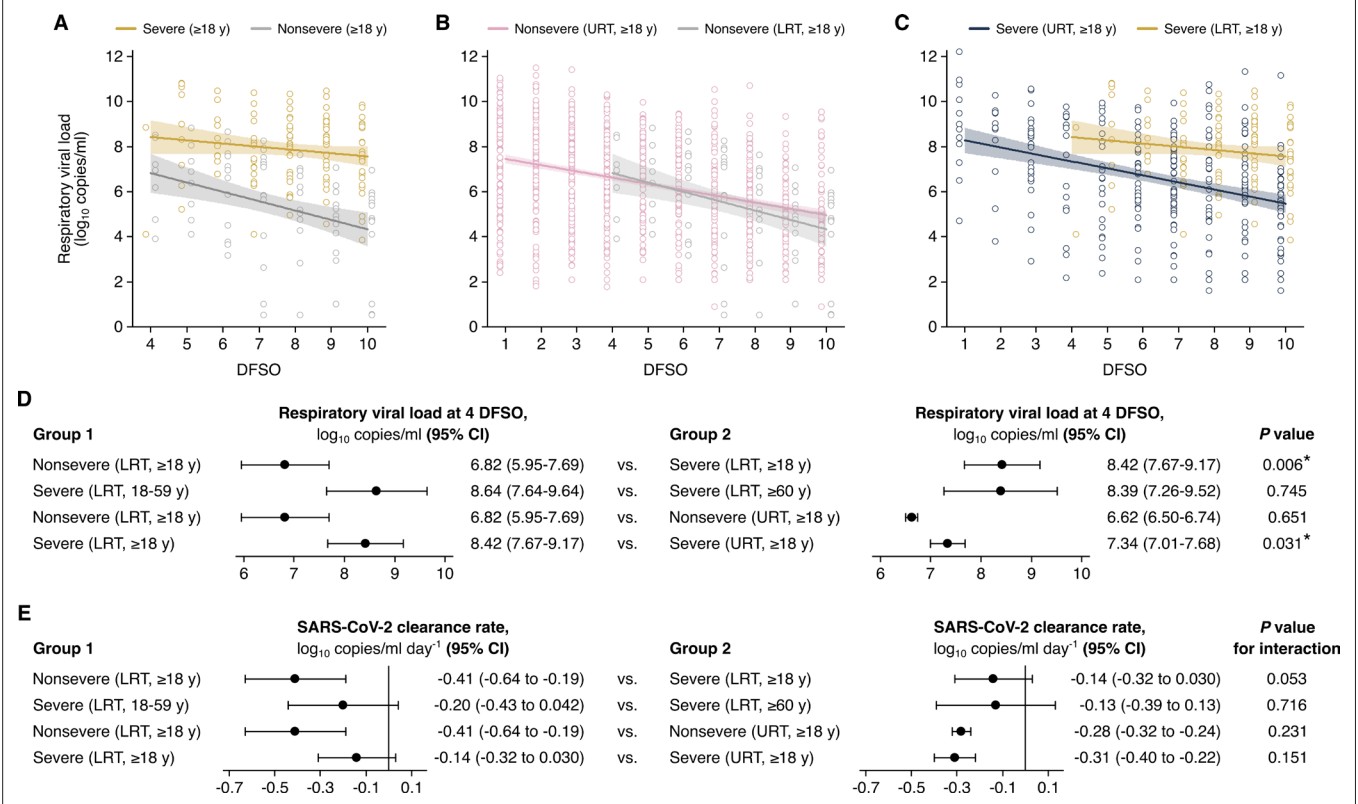

**Figure 3.** Comparison of severe acute respiratory syndrome coronavirus 2 (SARS-CoV-2) shedding in the lower respiratory tract (LRT) across severity and age groups and the upper respiratory tract (URT). (**A–C**). Shedding in the LRT for severe and nonsevere adult (aged 18 years or older) COVID-19 (**A**), in the LRT and URT nonsevere adult COVID-19 (**B**) and in the LRT and URT severe adult COVID-19 (**C**). Open circles represent respiratory viral load (rVL) data and were offset from their day from symptom onset (DFSO) for visualization. Lines and bands show regressions and their 95 % CIs, respectively. (**D and E**) Comparisons of shedding levels at 4 DFSO (**D**) and URT shedding dynamics (**E**) between severity and age groups in the LRT and between the LRT and URT. The black line in (**E**) depicts 0, the threshold for no significant trend in SARS-CoV-2 clearance. Linear regression analyses with interaction determined p-values and compared shedding levels and dynamics between the two groups in each row.

The online version of this article includes the following figure supplement(s) for figure 3:

**Figure supplement 1.** Severe acute respiratory syndrome coronavirus 2 (SARS-CoV-2) shedding in the lower respiratory tract (LRT) for adult coronavirus disease 2019 (COVID-19).

Between pediatric cases, who had nonsevere illness in our dataset, and adults with nonsevere illness, both URT shedding at 1 DFSO (p = 0.653) and URT dynamics (p for interaction = 0.400) were similar (*Figure 2B, F and G*). In contrast, URT shedding at 1 DFSO was greater for severely affected adults when compared to children with nonsevere disease (p = 0.017), but URT dynamics remained similar (p for interaction = 0.863) (*Figure 2C, F and G*).

## LRT shedding of SARS-CoV-2 for adult COVID-19

For adults, our analyses showed that high, persistent LRT shedding of SARS-CoV-2 was associated with severe COVID-19 but not nonsevere illness (*Figure 3A*). At the initial day in our analyzed period (4 DFSO), the mean rVL in the LRT of severe cases (8.42 [95% CI: 7.67–9.17] $\log_{10}$ copies/ml) was significantly greater (p = 0.006) than that of nonsevere cases (6.82 [95% CI: 5.95–7.69] $\log_{10}$ copies/ml) (*Figure 3D*). Between severities, the difference in LRT clearance rates was marginally above the threshold for statistical significance (p for interaction = 0.053). Nonetheless, severe cases had persistent LRT shedding, with no significant trend in SARS-CoV-2 clearance up to 10 DFSO (–0.14 [95% CI: –0.32–0.030] $\log_{10}$ copies/ml/day, p = 0.105), whereas nonsevere cases rapidly cleared the virus from the LRT (–0.41 [95% CI: –0.64 to –0.19] $\log_{10}$ copies/ml/day, p < 0.001) (*Figure 3E*). For nonsevere cases, the estimated mean duration of LRT shedding (down to 0 $\log_{10}$ copies/ml) was 20.4 (95% CI: 13.2–27.7) DFSO.

For severe COVID-19, regression analysis showed comparable mean LRT rVLs at 4 DFSO between younger and older adults (p = 0.745) (*Figure 3D*). For severe cases, both age groups also had persistent LRT shedding in the analyzed period: younger adults (–0.20 [95% CI: –0.32 to 0.042] $\log_{10}$ copies/ml/day, p = 0.105) and older adults (–0.13 [95% CI: –0.39 to 0.13] $\log_{10}$ copies/ml/day, p = 0.316) both had no significant trend in SARS-CoV-2 clearance (*Figure 3E*, *Figure 3—figure supplement 1A*). Likewise, severely affected male cases had no significant trend in LRT shedding (0.001 [95% CI: –0.16 to 0.19] $\log_{10}$ copies/ml/day, p = 0.988) (*Figure 3—figure supplement 1B*). The female group included few samples, and statistically analyses were not conducted (*Appendix 1—table 3*).

Interestingly, nonsevere cases showed similar SARS-CoV-2 shedding between the URT and LRT (*Figure 3B*), whereas severe cases shed greater and longer in the LRT than in the URT (*Figure 3C*). At 4 DFSO, the URT rVL of nonsevere adults was 6.62 (95% CI: 6.50–6.74) $\log_{10}$ copies/ml, which was not different from the LRT rVL of nonsevere adults (p = 0.651). Conversely, for severe adults, the rVL at 4 DFSO was significantly lower in the URT (7.34 [95% CI: 7.01–7.68] $\log_{10}$ copies/ml) than the LRT (p = 0.031) (*Figure 3D*).

## Heterogeneity in URT shedding of SARS-CoV-2

While regression analyses compared mean shedding levels and dynamics, we fitted rVLs to Weibull distributions to assess the heterogeneity in rVL. Both severe and nonsevere adult COVID-19 showed broad variation in URT shedding throughout disease course (*Figure 4A*). For severe disease, the standard deviation (SD) of rVL was 1.86, 2.34, 1.89, and 1.90 $\log_{10}$ copies/ml at 2, 4, 7, and 10 DFSO, respectively. For nonsevere illness, these SDs were 2.08, 1.90, 1.89, and 1.96 $\log_{10}$ copies/ml, respectively. Notably, our distribution analyses indicated that the top 2–9% of adults with COVID-19 harbored 80 % of the SARS-CoV-2 copies in the URT on each DFSO (*Figure 4—figure supplement 1A-D*).

Since cases with severe COVID-19 tend to deteriorate at 10 DFSO (*Solomon et al., 2020*; *Zhou et al., 2020*), early differences in shedding may predict disease severity before deterioration. To assess the prognostic utility of URT shedding, we used the rVL distributions of nonsevere and severe adult cases and calculated the AUC that is overlapped or separated (*Figure 4B*). The greater the separation between these rVL distributions, the greater the ability to differentiate severe COVID-19 from nonsevere illness, and this AUC analysis estimates the maximal accuracy of prognostication (*Figure 4C*). At each DFSO, these URT distributions were largely overlapped. Moreover, the cumulative density distributions of rVL (*Figure 4D*) estimated poor sensitivity and specificity for prognostication (*Figure 4—figure supplement 2*). Thus, our data indicated that URT shedding inaccurately predicts COVID-19 severity.

## Heterogeneity in LRT shedding of SARS-CoV-2

In contrast, the distributions of severe and nonsevere LRT shedding bifurcated along disease course (*Figure 4E*). At 6 DFSO, the estimate at the 80th case percentile of LRT rVL was 9.40 (95% CI: 8.67–10.20) $\log_{10}$ copies/ml for severe COVID-19, while it was 7.66 (95% CI: 6.65–8.83) $\log_{10}$ copies/ml for nonsevere illness. At 10 DFSO, the difference between 80th case percentile estimates expanded, as they were 8.63 (95% CI: 8.04–9.26) and 6.01 (95% CI: 4.65–7.78) $\log_{10}$ copies/ml for severe and nonsevere disease, respectively. Furthermore, our data indicated that nonsevere illness was associated with greater skewing in LRT shedding than severe disease in the analyzed period (*Figure 4E*). For nonsevere COVID-19, the SD of rVL was 1.92, 2.01, and 2.09 $\log_{10}$ copies/ml at 6, 8, and 10 DFSO, respectively. For severe disease, it was lesser at 1.25, 1.37, and 1.61 $\log_{10}$ copies/ml for 6, 8, and 10 DFSO, respectively. On each DFSO, the top 2–12% of cases harbored 80 % of the LRT copies of SARS-CoV-2 for adults with nonsevere COVID-19, whereas it was the top 10–20% of cases for adults with severe disease (*Figure 4—figure supplement 1, E to H*).

We also assessed the prognostic utility of LRT shedding. We calculated the AUC that is overlapped or separated, which showed greater separation between the LRT distributions of severe and nonsevere cases (*Figure 4F*). The estimated accuracy for using LRT shedding as a prognostic indicator for COVID-19 severity was up to 81 % (*Figure 4G*). As a resource, the cumulative distributions of LRT shedding (*Figure 4H*) enable for the estimation of the specificity and sensitivity at different prognostic thresholds of LRT rVL. For example, at 5 DFSO, the estimated specificity was 93.3 % and the estimated sensitivity was 64.4 % at a prognostic threshold of 9.10 $\log_{10}$ copies/ml (*Figure 4—figure supplement 3*). For 8 DFSO, the estimated specificity and sensitivity was 73.1% and 88.8%, respectively, at a

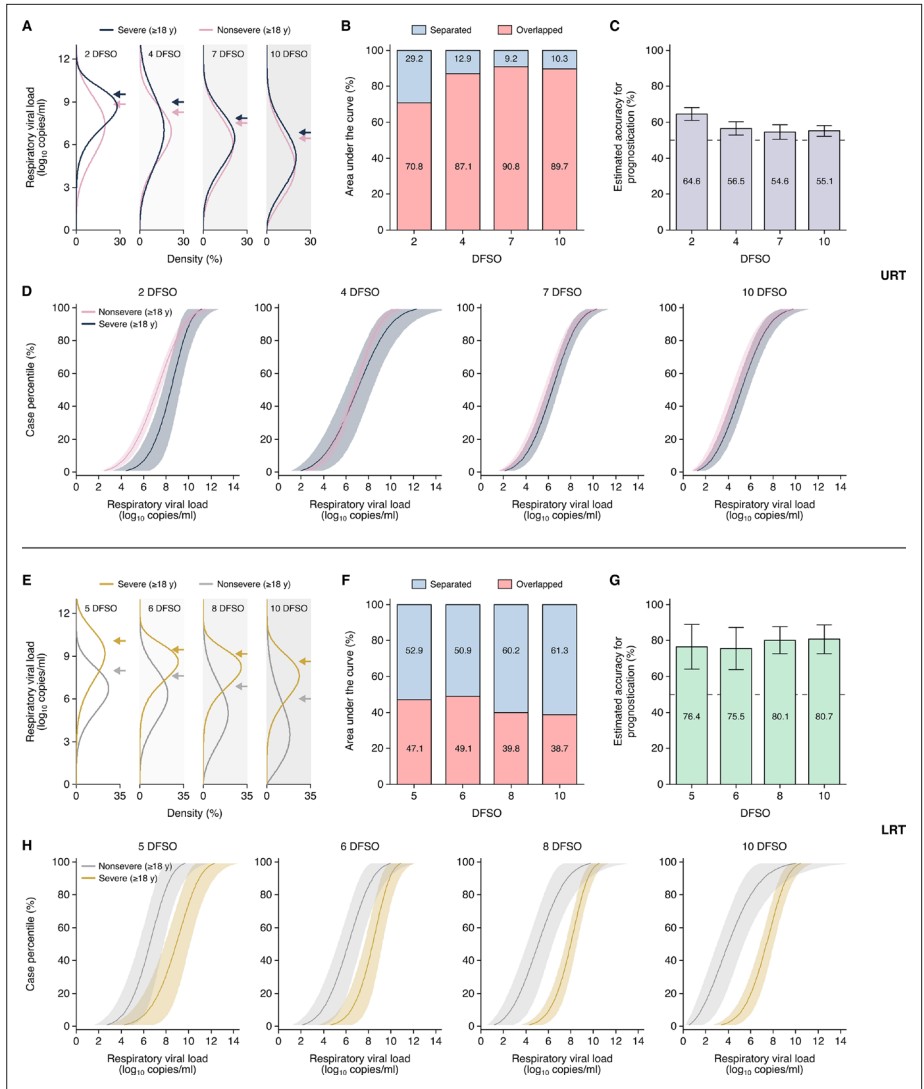

**Figure 4.** Heterogeneity in, and severity prognostication from, severe acute respiratory syndrome coronavirus 2 (SARS-CoV-2) shedding for adult coronavirus disease 2019 (COVID-19). (**A–D**) Upper respiratory tract (URT) analyses. (**A**) Estimated distributions at 2, 4, 7, and 10 days from symptom onset (DFSO) of URT shedding for adults (aged 18 years or older) with nonsevere or severe COVID-19. (**B**) Overlapped or separated areas under the curve for the distributions in (**A**). (**C**) Estimated accuracy for using URT shedding of SARS-CoV-2 as a prognostic indicator for COVID-19 severity. (**D**) Cumulative distributions of URT shedding for adults with nonsevere or severe COVID-19 at various DFSO. (**E–H**) Lower respiratory tract (LRT) analyses. (**E**) Estimated distributions at 5, 6, 8, and 10 DFSO of LRT shedding for adults with nonsevere or severe COVID-19. (**F**) Overlapped or separated areas under the curve for the distributions in (**E**). (**G**) Estimated accuracy for using LRT shedding of SARS-CoV-2 as a prognostic indicator for COVID-19 severity. (**H**) Cumulative distributions of LRT shedding for adults with nonsevere or severe COVID-19 at various DFSO. Arrows in (**A**) and (**E**) denote the 80th case percentiles, in terms of respiratory viral load (rVL), for each group. For (**D**) and (**H**), the proportion of cases to the left of a given prognostic threshold are predicted to have nonsevere COVID-19, while those to the right of it are predicted to have severe disease. Sensitivity and specificity can then be estimated using the nonsevere and severe distributions. The dotted lines in (**D**) and (**H**) denote 50 % accuracy.

The online version of this article includes the following figure supplement(s) for figure 4:

**Figure supplement 1.** Few cases carry the majority of severe acute respiratory syndrome coronavirus 2 (SARS-CoV-2) copies in the upper respiratory tract (URT) and lower respiratory tract (LRT).

**Figure supplement 2.** Estimated sensitivity and specificity of upper respiratory tract (URT) shedding as a prognostic indicator for severe acute respiratory syndrome coronavirus 2 (SARS-CoV-2) infection.

**Figure supplement 3.** Estimated sensitivity and specificity of lower respiratory tract (LRT) shedding as a prognostic indicator for severe acute respiratory syndrome coronavirus 2 (SARS-CoV-2) infection.

prognostic threshold of 5.95 $\log_{10}$ copies/ml. These estimated specificities and sensitivities agreed with the estimated accuracy for prognostication from their AUC analyses. Taken together, our data indicated that LRT shedding more accurately predicts COVID-19 severity than does URT shedding.

## Discussion

Our study systematically developed a dataset of COVID-19 case characteristics and rVLs and conducted stratified analyses on SARS-CoV-2 shedding post-symptom onset. In the URT, we found that adults with severe COVID-19 showed slightly higher rVLs shortly after symptom onset, but similar SARS-CoV-2 clearance rates, when compared with their nonsevere counterparts. After stratifying for disease severity, our analyses showed that sex and age had nonsignificant effects on SARS-CoV-2 shedding for each included analysis (summarized in *Appendix 1—table 4*). Thus, while sex and age influence the tendency to develop severe COVID-19 (*Onder et al., 2020*; *Tartof et al., 2020*; *Zhou et al., 2020*), we find no such sex dimorphism or age distinction in shedding among cases of similar severity. This includes children, who had nonsevere illness in our study and show similar URT shedding post-symptom onset as adults with nonsevere illness.

Notably, our analyses indicate that high, persistent LRT shedding of SARS-CoV-2 characterizes severe COVID-19 in adults. Previous reports have found prolonged LRT shedding for weeks in critically ill adult patients (*Buetti et al., 2020*; *Huang et al., 2020*). Our results provide additional insights into the LRT kinetics of SARS-CoV-2 in adults, particularly soon after symptom onset. They reveal a severity-associated difference in both shedding and clearance in the LRT which begins, at least, at 4 DFSO; our dataset had limited LRT samples before 4 DFSO. Interestingly, our analyses also reveal an early bifurcation between the LRT and URT for severe COVID-19. That is, severe disease is associated with higher rVLs in the LRT than the URT throughout the analyzed period, whereas nonsevere illness shows similar shedding between the LRT and URT. This suggests that the effective immune responses associated with milder COVID-19, including innate, cross-reactive, and coordinated adaptive immunity (*Lucas et al., 2020*; *Rydyznski Moderbacher et al., 2020*; *Ng et al., 2020*; *Pierce et al., 2020*; *Takahashi et al., 2020*), do not significantly inhibit early, or prolonged, SARS-CoV-2 replication in the LRT of severely affected adults. Hence, poorly controlled LRT replication tends to continue, at least, to 10 DFSO, which coincides with the timing of clinical deterioration (median, 10 DFSO) (*Solomon et al., 2020*; *Zhou et al., 2020*). Moreover, the bifurcated profiles of LRT shedding concur with the observed severity-associated differences in lung pathology, in which severe cases show hyperinflammation and progressive loss of epithelial-endothelial integrity (*Magro et al., 2020*; *Matheson and Lehner, 2020*; *Xu et al., 2020b*).

Thus, LRT shedding may predict COVID-19 severity, serving as a prognostic factor. As emerging evidence suggests that timing influences the efficacy of anti-SARS-CoV-2 therapies (*O'Brien et al., 2021*; *Weinreich et al., 2021a*), early clinical decision making is crucial. A prognostic indicator guides early risk stratification, identifying high-risk individuals before they deteriorate into severe COVID-19. This facilitates the early administration of the efficacious therapies to these patients and may reduce the incidence of severe and fatal COVID-19 (*O'Brien et al., 2021*; *Weinreich et al., 2021a*; *Weinreich et al., 2021b*). Additional studies should further explore the prognostic utility of LRT shedding in clinical settings, including toward improving COVID-19 outcomes.

LRT shedding can be assessed noninvasively. This study predominantly analyzed expectorated sputum, which can be obtained from a deep cough, as the LRT specimen. Since SARS-CoV-2 detection occurs more frequently in expectorated sputum than in URT specimens, including nasopharyngeal swabs (*Fajnzylber et al., 2020*; *Wang et al., 2020*; *Wölfel et al., 2020*), SARS-CoV-2 quantitation from sputum may more accurately diagnose COVID-19 while simultaneously predicting severity. Noninvasively induced sputum presents a potential alternative for patients without sputum production (*Lai et al., 2020*), although it was not assessed in this study and its prognostic utility remains to be evaluated. Furthermore, our data suggest that sex and age may not significantly influence prognostic thresholds but that the time course of disease may. Prognostication should account for the dynamics of shedding, and both the rVL and DFSO of a sputum specimen should be considered.

While our analyses did not account for virus infectivity, higher SARS-CoV-2 rVL is associated with a higher likelihood of culture positivity, from adults (*van Kampen et al., 2021*; *Wölfel et al., 2020*) as well as children (*L'Huillier et al., 2020*), and a higher transmission risk (*Marks et al., 2021*). Hence, our results suggest that infectiousness increases with COVID-19 severity, concurring with epidemiological

analyses (*Li et al., 2021*; *Sayampanathan et al., 2021*). They also suggest that adult and pediatric infections of similar severity have comparable infectiousness, reflecting epidemiological findings on age-based infectiousness (*Laxminarayan et al., 2020*; *Li et al., 2021*; *Sun et al., 2021*). Furthermore, since respiratory aerosols are typically produced from the LRT (*Johnson et al., 2011*), severe SARS-CoV-2 infections may have increased, and extended, risk for aerosol transmission. As severe cases tend to be hospitalized, this provides one possible explanation for the elevated risk of COVID-19 among healthcare workers in inpatient settings (*Nguyen et al., 2020*); airborne precautions, such as the use of N95 or air-purifying respirators, should be implemented around patients with COVID-19.

Our study has limitations. First, while our study design systematically developed a large, diverse dataset, there were few severe female cases with LRT specimens and no severe pediatric cases included. Statistical comparisons involving these groups were not conducted. Additional studies should permit these remaining comparisons. Second, our analyses did not account for additional case characteristics, including comorbidities, and their relationships with SARS-CoV-2 kinetics remain unclear. Third, the review found that expectorated sputum was the predominant LRT specimen used for SARS-CoV-2 quantitation, and our analyses on LRT kinetics may not generalize to cases without sputum production. The systematic dataset also consisted largely of hospitalized patients, and our results may not generalize to asymptomatic infections.

In summary, our findings provide insight into the kinetics of SARS-CoV-2 and describe virological factors that facilitate the pathogenesis of severe COVID-19. They show that high, persistent LRT shedding characterizes severe disease in adults, highlighting the potential prognostic utility of SARS-CoV-2 quantitation from LRT specimens. Lastly, each study identified by our systematic review collected specimens before October 2020. As widespread transmission of the emerging variants of concerns likely occurred after this date (*Davies et al., 2021*; *Konings et al., 2021*; *Tegally et al., 2021*), our study presents a quantitative resource to assess the effects of their mutations on respiratory shedding levels and dynamics.

## Acknowledgements

The authors thank S Fafi-Kremer, PharmD, PhD (Strasbourg University Hospital); Y Hirotsu, PhD (Yamanashi Central Hospital); MS Kelly, MD, MPH (Duke University); E Lavezzo, PhD, and A Crisanti, MD, PhD (University of Padova); JZ Li, MD, MMSc (Brigham & Women's Hospital); Cédric Laouénan, MD, PhD, and Yazdan Yazdanpanah, MD, PhD (Bichat-Claude Bernard University Hospital); NK Shrestha, MD (Cleveland Clinic); T Teshima, MD, PhD (Hokkaido University); S Trouillet-Assant, PhD (Université Hospital of Lyon); JJA van Kampen, MD, PhD (Erasmus University Medical Center); A Wyllie, PhD, N Grubaugh, PhD, and A Ko, MD (Yale School of Public Health); and A Yilmaz, MD, PhD (Sahlgrenska University Hospital) for responses to data inquiries.

## Additional information

### Competing interests

David N Fisman: reports serving on advisory boards of Seqirus, Sanofi Pasteur, Pfizer, and AstraZeneca, and consulting for the Ontario Nurses Association, Elementary Teachers' Federation of Ontario, JP MorganChase, WE Foundation, and Farallon Capital, outside the submitted work.. The other authors declare that no competing interests exist.

### Funding

| Funder | Grant reference number | Author |
|---|---|---|
| Natural Sciences and Engineering Research Council of Canada | Vanier Scholarship (608544) | Paul Z Chen |
| Canadian Institutes of Health Research | Canadian COVID-19 Rapid Research Fund (OV4-170360) | David N Fisman |

| Funder | Grant reference number | Author |
|---|---|---|
| Natural Sciences and Engineering Research Council of Canada | Senior Industrial Research Chair | Frank X Gu |
| Toronto COVID-19 Action Fund | | Frank X Gu |
| World Health Organization | | Niklas Bobrovitz |
| Public Health Agency of Canada | COVID-19 Immunity Task Force | Niklas Bobrovitz |

The funders had no role in study design, data collection and interpretation, or the decision to submit the work for publication.

## Author contributions

Paul Z Chen, Conceptualization, Formal analysis, Investigation, Methodology, Validation, Visualization, Writing – original draft, Writing – review and editing; Niklas Bobrovitz, Formal analysis, Investigation, Methodology, Writing – review and editing; Zahra A Premji, Methodology, Resources, Writing – review and editing; Marion Koopmans, Supervision, Writing – review and editing; David N Fisman, Methodology, Supervision, Writing – review and editing; Frank X Gu, Funding acquisition, Supervision, Writing – review and editing

## Author ORCIDs

Paul Z Chen http://orcid.org/0000-0001-5261-1610
Niklas Bobrovitz http://orcid.org/0000-0001-7883-4484
Zahra A Premji http://orcid.org/0000-0002-6899-0528
Marion Koopmans http://orcid.org/0000-0002-5204-2312
David N Fisman http://orcid.org/0000-0001-5009-6926
Frank X Gu http://orcid.org/0000-0001-8749-9075

## Decision letter and Author response

Decision letter https://doi.org/10.7554/eLife.70458.sa1
Author response https://doi.org/10.7554/eLife.70458.sa2

# Additional files

## Supplementary files
• Transparent reporting form

## Data availability

The systematic dataset and model outputs from this study can be download from a public repository (https://zenodo.org/record/5209064). The code generated during this study is available at GitHub (https://github.com/paulzchen/sars2-shedding; copy archived at https://archive.softwareheritage.org/swh:1:rev:c96390f98f47f17939f3669c7c8fad96f9603e84). The systematic review protocol was prospectively registered on PROSPERO (registration number, CRD42020204637).

The following dataset was generated:

| Author(s) | Year | Dataset title | Dataset URL | Database and Identifier |
|---|---|---|---|---|
| Chen PZ, Bobrovitz N, Premji Z, Koopmans M, Fisman DN, Frank XG | 2021 | SARS-CoV-2 viral loads across the upper and lower respiratory tract, sex, disease severity and age groups for adult and pediatric COVID-19 | https://zenodo.org/record/5209064 | Zenodo, 10.5281/zenodo.5209064 |

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

## Appendix 1

**Appendix 1—table 1.** Characteristics of contributing studies.

| Study | Country | Case types* | No. of adult cases (nonsevere; severe) | No. of pediatric cases (nonsevere; severe) | URT specimens (ascertained† for age, sex, disease severity) | LRT specimens (ascertained† for age, sex, disease severity) | Reported treatment (type)‡ | Volume of VTM reported | Specimen type (adjusted rVL)§ | Risk of bias¶ |
|---|---|---|---|---|---|---|---|---|---|---|
| **Bal et al., 2020** | France | H, N, A, S | 39 (38; 1) | 0 | 47 (47; 47; 47) | 0 | N/A | No | NPS (yes) | ***** |
| **Benotmane et al., 2020** | France | H, A, S | 37 (19; 18) | 0 | 47 (47; 47; 47) | 0 | Yes (azithromycin, other antibiotics, azole antifungals, lopinavir-ritonavir, hydroxychloroquine, tocilizumab, high-dose corticosteroid, mycophenolate mofetil withdrawal, mycophenolic acid withdrawal, calcineurin inhibitors withdrawal, mammalian target of rapamycin withdrawal, delayed belatacept administration) | No | NPS (yes) | ****** |
| **Biguenet et al., 2021** | France | H, N, A, S | 453 (406; 47) | 0 | 453 (453; 453; 453) | 0 | Yes (antibiotics, oseltamivir, hydroxychloroquine, corticosteroids, lopinavir-ritonavir, remdesivir) | No | NPS (yes) | ******** |
| **Fajnzylber et al., 2020** | USA | H, N, A, S | 25 (9; 16) | 0 | 31 (31; 31; 31) | 11 (11; 11) | Yes (remdesivir) | Yes | (NPS (yes), OPS (yes), Spu (yes)) | ******** |
| **Han et al., 2020** | South Korea | H, P, S, As | 0 | 12 (12; 0) | 33 (33; 10; 33) | 0 | N/A | No | NPS (yes) | ******** |
| **Hirotsu et al., 2020** | Japan | H, A, S | 3 (3; 0) | 0 | 9 (9; 9; 9) | 0 | N/A | No | NPS (yes) | **** |
| **Hurst et al., 2020** | USA | C, A, P, S, Ps | 17 (17; 0) | 76 (76; 0) | 93 (93; 93; 93) | 0 | Yes (remdesivir) | No | NPS (yes) | ******** |
| **Iwasaki et al., 2020** | Japan | H, A, S | 5 (4; 1) | 0 | 5 (5; 5; 5) | 0 | N/A | No | NPS (yes) | **** |
| **L'Huillier et al., 2020** | Switzerland | H, P, S | 0 | 23 (23; 0) | 23 (23; 0; 23) | 0 | N/A | No | NPS (yes) | ******** |
| **Lavezzo et al., 2020** | Italy | H, C, A, P, S, Ps | 40 (34; 6) | 2 (2; 0) | 6 (69; 69; 69) | 0 | N/A | Yes | NPS (yes), OPS (yes) | ******* |
| **Pan et al., 2020** | China | H, A, S, Ps | 2 (2; 0) | 0 | 13 (0; 0; 13) | 12 (0; 12) | N/A | No | OPS (yes), Spu (no) | **** |
| **Peng et al., 2020** | China | H, A, S | 6 (6; 0) | 0 | 6 (6; 6; 6) | 0 | Yes (arbidol, lopinavir, ritonavir, interferon alfa-2b inhalation) | No | OPS (yes) | ******** |
| **Shrestha et al., 2020** | USA | N, A, S | 196 (196; 0) | 1 (1; 0) | 213 (213; 213; 213) | 0 | Yes (indicated no hydroxychloroquine or other COVID-19-related treatments were used) | No | NPS (yes) | ******* |
| **Sun et al., 2020** | China | H, A, S | 6 (3; 3) | 0 | 6 (0; 0; 6) | 0 | N/A | No | NPS (yes), OPS (yes), Spu (no) | ***** |
| **To et al., 2020** | China | H, A, S | 4 (2; 2) | 0 | 2 (0; 0; 2) | 2 (0; 2) | N/A | Yes | ETA (yes), POS (yes) | ********* |
| **van Kampen et al., 2021** | The Netherlands | H, A, S | 171 (8; 163) | 0 | 57 (57; 57; 57) | 114 (114, 114) | Yes (lopinavir–ritonavir with or without ribavirin or interferon beta 1b) | Yes | NPS (yes), Spu (yes) | ******** |
| **Vetter et al., 2020** | Switzerland | H, A, S | 5 (4; 1) | 0 | 63 (63; 63; 63) | 0 | Yes (paracetamol, alfuzosin, ibuprofen, enoxaparin, amoxicillin clarithromycin, piperacillin, tazobactam, lopinavir, ritonavir, folic acid) | Yes | NPS (yes), OPS (yes) | ********* |
| **Wölfel et al., 2020** | Germany | H, A, S, As | 9 (9; 0) | 0 | 71 (71; 0; 71) | 70 (70; 0; 70) | N/A | Yes | NPS (yes), OPS (yes), Spu (no) | ******* |
| Wyllie et al. | USA | H, A, P, S | 31 (31; 0) | 2 (2; 0) | 33 (0; 0; 33) | 0 | N/A | Yes | NPS (yes) | ******* |
| **Xu et al., 2020a** | China | H, P, S, Ps, As | 0 | 9 (9; 0) | 39 (39; 39; 39) | 0 | Yes (α-interferon oral spray, azithromycin) | No | NPS (yes) | ******** |

*Appendix 1—table 1 Continued on next page*

Appendix 1—table 1 Continued

| Study | Country | Case types[*] | No. of adult cases (nonsevere; severe) | No. of pediatric cases (nonsevere; severe) | URT specimens (ascertained[†] for age, sex, disease severity) | LRT specimens (ascertained[†] for age, sex, disease severity) | Reported treatment (type)[‡] | Volume of VTM reported | Specimen type (adjusted rVL)[§] | Risk of bias[¶] |
|---|---|---|---|---|---|---|---|---|---|---|
| **Yazdanpanah, 2021** | France | H, A, Ps, S | 125 (64; 61) | 0 | 173 (173; 173; 173) | 0 | Yes (remdesivir, hydroxychloroquine, lopinavir-ritonavir) | No | NPS (yes) | ******** |
| **Yilmaz et al., 2021** | Sweden | H, A, S | 52 (37; 12) | 0 | 102 (102; 102; 102) | 0 | N/A | Yes | NPS (yes), OPS (yes) | ******* |
| **Yonker et al., 2020** | USA | H, A, P, S | 3 (3; 0) | 14 (13; 1) | 14 (14; 0; 14) | 0 | N/A | No | NPS (yes) | ****** |
| **Zhang et al., 2021** | China | H, A, S | 4 (3; 1) | 0 | 10 (10; 10; 10) | 0 | N/A | No | NPS (yes), OPS (yes) | ******** |
| **Zheng et al., 2020** | China | H, A, S | 17 (10; 7) | 0 | 13 (13; 0; 13) | 12 (12; 0; 12) | Yes (gammaglobulin, glucocorticoids, antibiotics, antiviral combination of interferon α inhalation, lopinavir-ritonavir combination, arbidol, favipiravir, and darunavir-cobicistat) | No | POS (yes), Spu (yes) | ******* |
| **Zou et al., 2020** | China | H, A, S, As | 16 (13; 3) | 0 | 88 (88; 88; 88) | 0 | N/A | No | NPS (yes), OPS (yes) | ******* |

[*]Hospitalized cases were those admitted to a hospital. Non-admitted cases were those tested in a hospital setting but not admitted. Community cases were those tested in a community setting.

[†]The number of specimens for which their cases were ascertained for age group (aged 0–17 years, 18–59 years, or 60 years or older), sex (male or female), or disease severity (nonsevere or severe) via data reported in the study or via data request. All specimens were ascertained for the DFSO on which they were taken.

[‡]Responses of 'N/A' indicate that no details were reported on treatment for COVID-19 in the study.

[§]Specimen measurements were converted to rVLs based on the reported, or assumed, dilution factor for specimens immersed in transport media.

[¶]The modified Joanna Briggs Institute (JBI) critical appraisal checklist was used, with more stars indicating lower risk of bias. Studies were considered to have low risk of bias if they met the majority of the items (≥6/10 items) and met item 1. Results from each study are shown in Appendix—table 2.

H = hospitalized. N = non-admitted. C = community. A = adult (aged 18 years or older). P = pediatric (aged 0–17 years). S = symptomatic. Ps = presymptomatic. As = asymptomatic. rVL = respiratory viral load. endotracheal aspirate (ETA); nasopharyngeal swab (NPS); oropharyngeal swab (OPS); posterior oropharyngeal saliva (POS); and sputum (Spu).

**Appendix 1—table 2.** Assessment of risk of bias based on the modified Joanna Briggs Institute (JBI) critical appraisal checklist.

| Study | Checklist items* | | | | | | | | | |
|---|---|---|---|---|---|---|---|---|---|---|
| | 1 | 2 | 3 | 4 | 5 | 6 | 7 | 8 | 9 | 10 |
| *Bal et al., 2020* | N | Y | U | U | N | Y | Y | N | Y | Y |
| *Benotmane et al., 2020* | N | Y | Y | Y | N | Y | Y | N | Y | Y |
| *Biguenet et al., 2021* | Y | Y | Y | Y | N | Y | Y | N | Y | Y |
| *Fajnzylber et al., 2020* | Y | Y | Y | Y | N | Y | Y | Y | N | Y |
| *Han et al., 2020* | Y | Y | Y | Y | N | Y | Y | N | Y | Y |
| *Hirotsu et al., 2020* | Y | N | U | U | N | Y | Y | N | N | Y |
| *Hurst et al., 2020* | Y | Y | Y | Y | N | Y | Y | N | Y | Y |
| *Iwasaki et al., 2020* | Y | N | U | U | N | Y | Y | N | N | Y |
| *L'Huillier et al., 2020* | Y | Y | Y | Y | N | Y | Y | N | Y | Y |
| *Lavezzo et al., 2020* | Y | Y | N | Y | N | Y | Y | N | Y | Y |
| *Pan et al., 2020* | Y | N | U | U | N | Y | Y | N | N | Y |
| *Peng et al., 2020* | Y | Y | Y | Y | N | Y | Y | N | Y | Y |
| *Shrestha et al., 2020* | N | Y | Y | Y | N | Y | Y | N | Y | Y |
| *Sun et al., 2020* | N | N | Y | Y | N | Y | Y | N | N | Y |
| *To et al., 2020* | Y | Y | Y | Y | N | Y | Y | Y | Y | Y |
| *van Kampen et al., 2021* | Y | Y | Y | Y | N | Y | Y | N | Y | Y |
| *Vetter et al., 2020* | Y | Y | Y | Y | N | Y | Y | Y | Y | Y |
| Wölfel et al. (2020) | Y | Y | N | U | N | Y | Y | Y | Y | Y |
| *Wyllie et al., 2020* | Y | Y | U | Y | N | Y | Y | Y | N | Y |
| *Xu et al., 2020b* | Y | Y | Y | Y | N | Y | Y | N | Y | Y |
| *Yazdanpanah, 2021* | Y | Y | Y | Y | N | Y | Y | N | Y | Y |
| *Yilmaz et al., 2021* | Y | Y | U | U | N | Y | Y | Y | Y | Y |
| *Yonker et al., 2020* | Y | Y | N | U | N | Y | Y | N | Y | Y |
| *Zhang et al., 2021* | Y | Y | Y | Y | N | Y | Y | N | Y | Y |
| *Zheng et al., 2020* | Y | Y | Y | Y | N | Y | Y | N | N | Y |
| *Zou et al., 2020* | N | Y | Y | Y | N | Y | Y | N | Y | Y |

*Descriptions of each item are included in the modified JBI critical appraisal checklist. Grey, yellow, and red represent yes (Y), unclear (U), and no (N), respectively.

**Appendix 1—table 3.** Summary of respiratory shedding levels and dynamics for coronavirus disease 2019 (COVID-19) groups.

| Group | $n^{\dagger *}$ | rVL*† (95% CI), log₁₀ copies/ml | | SARS-CoV-2 clearance rate | |
|---|---|---|---|---|---|
| | | At 1 DFSO | At 4 DFSO | Estimate (95% CI), log₁₀ copies/ml/day | p-Value‡ |
| **URT, ≥18 years** | | | | | |
| Nonsevere | 1 092 | 7.45 (7.26–7.65) | 6.62 (6.50–6.74) | −0.28 (−0.32 to −0.24) | <0.001 |
| Severe | 289 | 8.28 (7.71–8.84) | 7.34 (7.01–7.68) | −0.31 (−0.40 to −0.22) | <0.001 |
| Male (nonsevere) | 382 | 7.66 (7.31–8.01) | – | −0.28 (−0.36 to −0.21) | <0.001 |
| Female (nonsevere) | 618 | 7.29 (7.05–7.54) | – | −0.26 (−0.32 to −0.21) | <0.001 |
| Male (severe) | 175 | 8.00 (7.23–8.78) | – | −0.27 (−0.39 to −0.15) | <0.001 |
| Female (severe) | 89 | 8.59 (7.65–9.54) | – | −0.38 (−0.54 to −0.22) | <0.001 |
| Nonsevere (18–59 years) | 857 | 7.53 (7.31–7.74) | – | −0.30 (−0.35 to −0.26) | <0.001 |
| Nonsevere (≥ 60 years) | 212 | 7.26 (6.77–7.74) | – | −0.22 (−0.31 to −0.12) | <0.001 |
| Severe (18–59 years) | 89 | 8.31 (7.28–9.33) | – | −0.37 (−0.54 to −0.21) | <0.001 |
| Severe (≥ 60 years) | 192 | 8.24 (7.54–8.93) | – | −0.28 (−0.39 to −0.16) | <0.001 |
| **LRT, ≥ 18 years§** | | | | | |
| Nonsevere | 80 | – | 6.82 (5.95–7.69) | −0.41 (−0.64 to −0.19) | <0.001 |
| Severe | 121 | – | 8.42 (7.67–9.17) | −0.14 (−0.32 to 0.030) | 0.105¶ |
| Male (severe) | 94 | – | 7.84 (7.03–8.65) | 0.001 (−0.16 to 0.19) | 0.988¶ |
| Severe (18–59 years) | 55 | – | 8.64 (7.64–9.64) | −0.20 (−0.43 to 0.042) | 0.105¶ |
| Severe (≥ 60 years) | 65 | – | 8.39 (7.26–9.52) | −0.13 (−0.39 to 0.13) | 0.316¶ |
| **URT, 0–17 years** | | | | | |
| Overall | 180 | 7.32 (6.78–7.86) | – | −0.32 (−0.42 to −0.22) | <0.001 |
| Male | 64 | 6.41 (5.52–7.31) | – | −0.18 (−0.33 to −0.022) | 0.026 |
| Female | 58 | 6.72 (5.59–7.85) | – | −0.30 (−0.49 to −0.10) | 0.004 |

*Respiratory viral loads (rVLs) were based on the regression estimate at 1 day from symptom onset (DFSO) for upper respiratory tract (URT) shedding or 4 DFSO for lower respiratory tract (LRT) shedding. Estimates at 4 DFSO were included for the nonsevere and severe URT (≥18 years) groups, as they were compared with their LRT counterparts.

†*n* represents the number of rVL samples per group (from 1 to 10 DFSO for URT shedding or 4 to 10 DFSO for LRT shedding).

‡p-Value for the clearance rate was based on the regression parameter (*t*-test).

§There was lower sample numbers in the nonsevere groups and female (LRT, severe, ≥18 years) group, and we did not include these analyses.

¶Non-significance (p > 0.05).

**Appendix 1—table 4.** Summary of statistical comparisons on severe acute respiratory syndrome coronavirus 2 (SARS-CoV-2) shedding, across the respiratory tract, coronavirus disease 2019 (COVID-19) severity, sex, and age groups.

| | | p-Value[*] | |
| --- | --- | --- | --- |
| Group 1 | Group 2 | Main effect[†] | Interaction[‡] |
| **URT, ≥18 years** | | | |
| Nonsevere | Severe | 0.005* | 0.479 |
| Female (nonsevere) | Male (nonsevere) | 0.085 | 0.644 |
| Female (severe) | Male (severe) | 0.326 | 0.280 |
| Nonsevere (18–59 years) | Nonsevere ( ≥ 60 years) | 0.294 | 0.100 |
| Severe (18–59 years) | Severe ( ≥ 60 years) | 0.915 | 0.359 |
| **LRT, ≥18 years§** | | | |
| Nonsevere | Severe | 0.006* | 0.053 |
| Severe (18–59 years) | Severe ( ≥ 60 years) | 0.745 | 0.716 |
| **URT vs. LRT, ≥18 years** | | | |
| Nonsevere (URT, ≥18 years) | Nonsevere (LRT, ≥18 years) | 0.651 | 0.231 |
| Severe (URT, ≥18 years) | Severe (LRT, ≥18 years) | 0.031* | 0.151 |
| **URT, 0–17 years** | | | |
| Nonsevere (0–17 years) | Nonsevere (≥18 years) | 0.653 | 0.400 |
| Nonsevere (0–17 years) | Severe ( ≥ 18 years) | 0.017* | 0.863 |
| Female (nonsevere) | Male (nonsevere) | 0.667 | 0.333 |

[*]$p < 0.05$.

[†]p-Value for the main effect in linear regression analysis compares the mean respiratory viral loads (rVLs) at 1 day from symptom onset (DFSO) for the upper respiratory tract (URT) or, for any analyses including the lower respiratory tract (LRT), at 4 DFSO.

[‡]p-Value for interaction in linear regression analysis describes the difference in respiratory shedding dynamics along the time course of disease.

[§]There were small sample sizes in the nonsevere group and female (LRT, severe, ≥18 years) group, and these analyses were not included.

## Modified JBI critical appraisal checklist

| Reviewer | | | Date | | |
| --- | --- | --- | --- | --- | --- |
| Author | | | Year | Record Number | |
| | Yes | No | Unclear | Not applicable | |
| 1. Was the sample frame appropriate to address the target population? | ☐ | ☐ | ☐ | ☐ | |
| 2. Were the study subjects and the setting described in detail? | ☐ | ☐ | ☐ | ☐ | |
| 3. Did the study have consecutive inclusion of participants for case series and cohort studies? Did the study use probability-based sampling for cross-sectional studies? | ☐ | ☐ | ☐ | ☐ | |
| 4. Was the response rate adequate, and if not, was the low response rate managed appropriately? | ☐ | ☐ | ☐ | ☐ | |
| 5. Was the sample size adequate? | ☐ | ☐ | ☐ | ☐ | |
| 6. Were valid methods used for the identification of the condition? | ☐ | ☐ | ☐ | ☐ | |
| 7. Were standard, valid methods used for measurement of the exposure? | ☐ | ☐ | ☐ | ☐ | |
| 8. Was the exposure measured in an objective, reliable way for all participants? | ☐ | ☐ | ☐ | ☐ | |
| 9. Was there clear reporting of clinical information of the participants? | ☐ | ☐ | ☐ | ☐ | |
| 10. Was statistical analysis appropriate? | ☐ | ☐ | ☐ | ☐ | |

*Continued on next page*

*Continued*

| | | | | |
|---|---|---|---|---|
| **Reviewer** | | **Date** | | |
| **Author** | | **Year** | **Record Number** | |
| | **Yes** | **No** | **Unclear** | **Not applicable** |

Overall appraisal: Include ☐ Exclude ☐ Seek further info ☐

Comments (Including reason for exclusion)

## Tool Guidance

This modified checklist was based on a the JBI Critical Appraisal Checklists for case series, prevalence studies and analytical cross-sectional studies.

1. Was the sample frame appropriate to address the target population?
   This question relies upon knowledge of the broader characteristics of the population of interest and the geographical area.
   This study broadly investigates the respiratory viral load for the population of interest, which is the general population infected with SARS-CoV-2. The geographical area is not constrained. Sample frames restricted to particular subgroups within the general infected population were considered appropriate if they targeted one of the following groups analyzed in our study: asymptomatic, presymptomatic, symptomatic, adult, pediatric, hospitalized, non-admitted, or community.

2. Were the study subjects and the setting described in detail?
   Certain diseases or conditions vary in prevalence across different geographical regions and populations (e.g., women vs. men, sociodemographic variables between countries). The study sample should be described in sufficient detail so that other researchers can determine if it is comparable to the population of interest to them.

3. Did the study have consecutive inclusion of participants for case series and cohort studies? Did the study use probability sampling for cross-sectional studies?
   Inclusion of consecutive participants for case series and cohort studies yields results at lower risk of bias compared to other sampling methods for these study designs. Use of probability-based sampling methods for cross-sectional studies yields estimates at lower risk of bias compared to other sampling methods for this design. Studies that indicate a consecutive inclusion are more reliable than those that do not. For example, a case series that states 'we included all patients (24) with osteosarcoma who presented to our clinic between March 2005 and June 2006' is more reliable than a study that simply states 'we report a case series of 24 people with osteosarcoma'.

4. Was the response rate adequate, and if not, was the low response rate managed appropriately?
   A large number of dropouts, refusals or 'not founds' among selected subjects may diminish a study's validity, as can a low response rates for survey studies. The authors should clearly discuss the response rate and any reasons for non-response and compare persons in the study to those not in the study, particularly with regard to their sociodemographic characteristics. If reasons for non-response appear to be unrelated to the outcome measured and the characteristics of non-responders are comparable to those who do respond in the study, the researchers may be able to justify a more modest response rate.

5. Was the sample size adequate?
   The larger the sample, the narrower will be the confidence interval around the prevalence estimate, making the results more precise. An adequate sample size is important to ensure good precision of the final estimate. The sample size threshold was calculated as follows:

   $$n = \frac{z^2 \sigma}{d^2},$$

   where $n$ is the sample size threshold, $z$ is the z-score for the level of confidence (95%), $\sigma$ is the standard deviation (assumed to be 3 $\log_{10}$ copies/ml, a fourth of the full range of rVLs), and $d$ is the marginal error (assumed to be 1 $\log_{10}$ copies/ml, based on the minimum detection limit for qRT-PCR across studies). This item was met if ≥75 % of the included DFSO had ≥46 specimen measurements.

6. Were valid methods used for the identification of the condition?
Many health problems are not easily diagnosed or defined and some measures may not be capable of including or excluding appropriate levels or stages of the health problem. If the outcomes were assessed based on existing definitions or diagnostic criteria, then the answer to this question is likely to be yes. If the outcomes were assessed using observer reported, or self-reported scales, the risk of over- or under-reporting is increased, and objectivity is compromised. Importantly, determine if the measurement tools used were validated instruments as this has a significant impact on outcome assessment validity.

7. Were standard, valid methods used for measurement of the exposure?
The study should clearly describe the method of measurement of exposure. Assessing validity requires that a 'gold standard' is available to which the measure can be compared. The validity of exposure measurement usually relates to whether a current measure is appropriate or whether a measure of past exposure is needed.
In this study, standard methods to measure viral load in respiratory specimens are assays quantifying via one of the diagnostic sequences (*Ofr1b*, *N*, *RdRp,* and *E* genes) for SARS-CoV-2.

8. Was the exposure measured in an objective, reliable way for all participants?
The study should clearly describe the procedural aspects of the measurement of exposure as well as factors that can contribute to heterogeneity in measurement.
In this study, objective, reliable interpretation of the exposure depends on the use of quantitative calibration; the specification of extraction; determination of the viral load as a standard metric (e.g., copies/ml or equivalent) or in a manner that can be converted to a standard metric; and, if present, specification of the amount of diluent (e.g., viral transport media) used.

9. Was there clear reporting of clinical information of the participants?
There should be clear reporting of clinical information of the participants such as the following information where relevant: disease status, comorbidities, stage of disease, previous interventions/treatment, results of diagnostic tests, etc.
In addition, there should be clear reporting of the number and types (asymptomatic, presymptomatic, symptomatic, adult, pediatric, hospitalized, non-admitted, community, etc.) of cases for measurements within the sampling periods of interest. For studies that include data outside of the infectious period, there should be clear reporting of clinical information for participants for the specimen measurements that were collected from within the infectious period.

10. Was statistical analysis appropriate?

As with any consideration of statistical analysis, consideration should be given to whether there was a more appropriate alternate statistical method that could have been used. The methods section of studies should be detailed enough for reviewers to identify which analytical techniques were used and whether these were suitable.

| Risk of bias for each study | |
|---|---|
| Low | The majority of critical appraisal criteria are met (≥6/10 items) and included item 1 (representative sample). The estimates are likely to be correct for the target population. |
| High | The majority of critical appraisal criteria are not met (<6/10 items) or did not include item 1 (representative sample). This may impact on the validity and reliability of the estimates. The estimates may not be correct for the target population. |
| Unclear | The majority of items are unclear. There was insufficient information to assess the risk of bias. |

