## [Decision Letter]

**Acceptance summary:**

The authors performed a systematic literature review and meta-analysis to develop a dataset of respiratory viral loads (rVLs) for SARS-CoV-2. Focus was on finding the relation between individual case characteristics (e.g. disease severity, age and sex) and lower and upper respiratory tract viral loads. COVID-19 severity, rather than sex or age, predicts SARS-CoV-2 kinetics, and SARS-CoV-2 viral load from lower respiratory tract specimens seems to predict severe disease days before clinical deterioration for COVID-19 patients.

**Decision letter after peer review:**

Thank you for submitting your article "SARS-CoV-2 shedding dynamics across the respiratory tract, sex and disease severity for adult and pediatric COVID-19: a systematic review and modeling study" for consideration by *eLife*. Your article has been reviewed by 2 peer reviewers, and the evaluation has been overseen by Jos Van der Meer as the Senior and Reviewing Editor. The following individual involved in review of your submission has agreed to reveal their identity: Lucie Vermeulen (Reviewer #2).

Essential revisions:

1) It is not clear how surprising the association between LRT and disease severity is. For example, a similar result was found by Buetti et al., Critical Care, 2020 ("viral shedding in LRT lasted almost 30 days in median in critically ill patients, and the viral load in the LRT was associated with the 6-week mortality.") and the statement made in the methods of this paper ("In COVID-19 cases, rVL tends to diminish exponentially after 1 DFSO in the URT, whereas it tends to do so after 4 DFSO in the LRT (Bernheim et al., 2020; Chen et al., 2021; Wolfel et al., 2020)." ) also indicates that a faster decrease of viral load has been observed in URT compared to LRT. The present analysis certainly adds to these findings, but the observation does not seem to be completely new.

2) Lines 322-: "Data from serially sampled asymptomatic cases were included, and the day of laboratory diagnosis was referenced as 0 DFSO". It would be interesting to compare the dynamics observed in those serially sampled patients to the patterns shown in Figure 2 and derived mostly from cross-sectional data.

3) Lines 366-:"We used regression analysis to assess the respiratory shedding of SARS-CoV-2 and compare age, sex or severity groups. " Please specify the type of regression analysis (I guess this was "normal" linear regression assuming normally distributed erros). How are measurements with undetectable virus loads included in the analysis? Wouldn't these measurements call for the use of censored regression models?

4) The approach of how the authors estimated the AUC-ROC for the prediction of disease severity is not fully clear. Why did the authors need to first fit a distribution to the viral load values? The ROC curve and its AUC value can be computed directly from the observations, without this intermediate step.

5) Line 381-382: You write "Regression models were extrapolated (to 0 log10 copies/ml, rather than an assay detection limit) to estimate the duration of shedding." How large is the effect of extrapolating to 0 instead of to the detection limit? Some discussion on this is warranted.

6) Line 390: Why was the Weibull distribution chosen? Some reasoning for this could be added to the paper (in methods, discussion or supplement)

7) Line 399-400: "The fitted Weibull distributions were used to estimate the accuracy when using URT or LRT rVLs of SARS-CoV-2 as a prognostic indicator for SARS-CoV-2 infection." Do you mean severity of infection here?

8) Table 1: There are relatively few lower respiratory samples included in the analysis, compared to the number of upper respiratory samples. What is typically the reason that a LRT sample is taken from a patient? And following, is there any possible association of taking an LRT sample with patient characteristics included or excluded in this study that could influence the results? Then some discussion on this would be warranted.

9) Figure 4: Could it be made clearer in the figure which panels concern URT and which LRT? Now the reader has to carefully read the caption in order to deduce this.*Reviewer #1 (Recommendations for the authors):*

This manuscript presents a systematic review and regression analysis to analyze the association between upper and lower respiratory tract shedding (URT and LRT) of SARS-CoV-2 and disease severity. In addition, the authors study the impact of the days from symptoms onset on shedding in the two compartments. Overall, the presented results provide an interesting synthesis of the literature on these issues.

*Reviewer #2 (Recommendations for the authors):*

The study appears robust and comprehensive, and relevant quality checks for systematic review have been applied. The results are valuable and contribute to the scientific knowledge in this field.

Interesting findings include:

– Adult patients with severe disease had on average a somewhat higher upper respiratory tract viral load at 1 day from symptom onset than patients with non-severe disease. After this stratification for severity, respiratory viral loads did not differ significantly for age and sex. Rates of viral clearing were similar. Children and adults with non-severe disease had similar upper respiratory tract viral loads and viral clearance rates.

– High and persistent lower respiratory tract shedding of SARS-CoV-2 was associated with severe but not non-severe illness. The difference in lower respiratory viral load for severe and non-severe cases was more pronounced than for upper respiratory tract viral loads. In contrast to the upper respiratory tract, viral clearance from the lower respiratory tract was more rapid in non-severe than in severe cases. Again, age and sex did not differ significantly after stratification for severity.

– The authors then aimed to assess whether the observed difference in shedding in the first days after start of symptoms could be used to predict which people would develop more severe COVID-19. Typically, deterioration into severe disease only happens around 10 days from symptom onset. The authors conclude that upper respiratory tract viral shedding is so heterogeneous that its predictive capacity of disease severity is inaccurate. In contrast, lower respiratory tract shedding does have a predictive accuracy of up to 81% for disease severity.

Potential impact: Lower respiratory tract viral load could thus potentially be used as an early warning for developing severe COVID-19. However, lower respiratory tract samples are not routinely taken, the standard nasopharyngeal swab is an upper respiratory sample. Some discussion on the practical applicability of this suggestion could enhance the paper's impact.

---

## [Author Response]

Essential revisions:1) It is not clear how surprising the association between LRT and disease severity is. For example, a similar result was found by Buetti et al., Critical Care, 2020 ("viral shedding in LRT lasted almost 30 days in median in critically ill patients, and the viral load in the LRT was associated with the 6-week mortality.") and the statement made in the methods of this paper ("In COVID-19 cases, rVL tends to diminish exponentially after 1 DFSO in the URT, whereas it tends to do so after 4 DFSO in the LRT (Bernheim et al., 2020; Chen et al., 2021; Wolfel et al., 2020)." ) also indicates that a faster decrease of viral load has been observed in URT compared to LRT. The present analysis certainly adds to these findings, but the observation does not seem to be completely new.

We have included the reference and added discussion to contextualize our findings on LRT shedding: “Previous reports have found prolonged LRT shedding for weeks in critically ill adult patients (Buetti et al., 2020; Huang et al., 2020). Our results provide additional insights into the LRT kinetics of SARS-CoV-2 in adults, particularly soon after symptom onset. They reveal a severity-associated difference in both shedding and clearance in the LRT which begins, at least, at 4 DFSO; our dataset had limited LRT samples before 4 DFSO. Interestingly, our analyses also reveal an early bifurcation between the LRT and URT for severe COVID-19. That is, severe disease is associated with higher rVLs in the LRT than the URT throughout the analyzed period, whereas nonsevere illness shows similar shedding between the LRT and URT.” (page 11, line 295-303).

The referenced study (Buetti et al., Critical Care, 2020) showed an extended duration of LRT shedding for critically ill patients but did not find a statistical distinction in LRT viral load soon after symptom onset, perhaps due to limited sample numbers (see data in Figure 2 for <7 days and 7-14 days). Our analyses, perhaps due to greater sample numbers, found a statistical difference in LRT viral load levels (“main effect” at 4 DFSO) and dynamics (whether the rate of viral clearance was significant or not).

The quoted statement in the Methods was included to justify that viral load diminishes in, and we can apply a linear regression analysis to, the periods of 1-10 DFSO for URT shedding and 4-10 DFSO for LRT shedding. It was not intended to convey that LRT shedding peaks at 4 DFSO. LRT viral load may peak and diminish before 4 DFSO, although LRT viral load data near or before symptom onset is limited in this study and others. To better convey our intended meaning, we have amended this statement: “Previous studies have shown that SARS-CoV-2 shedding tends to diminish exponentially after 1 DFSO in the URT and, at least, after 4 DFSO in the LRT (Bernheim et al., 2020; P. Z. Chen et al., 2021; Wolfel et al., 2020). Although LRT shedding may peak before 4 DFSO, there is limited data near or before symptom onset.” (page 19, line 525-528).

2) Lines 322-: "Data from serially sampled asymptomatic cases were included, and the day of laboratory diagnosis was referenced as 0 DFSO". It would be interesting to compare the dynamics observed in those serially sampled patients to the patterns shown in Figure 2 and derived mostly from cross-sectional data.

We plotted the data for these cases and compared then from the regression lines, see Author response image 1:

**Author response image 1. sa2fig1:** Respiratory viral loads from serially sampled asymptomatic cases. Each set of connect dots represents a separate individual, except for one adult case which had paired NPS and Spu specimens. The data were overlaid on the regression lines for severe adult cases, nonsevere adult cases and nonsevere pediatric cases (Figure 2A-B). Data between 1-10 DFSO, as defined in the reviewer comment, is shown. The pediatric case with unchanged rVLs was below the detection limit for that study in the included DFSO. NPS, nasopharyngeal swab; Spu, sputum.

This comparison highlights the broad heterogeneity in SARS-CoV-2 shedding. As other studies have suggested, even asymptomatic infections show broad case variation in viral load. As this analysis did not have large sample numbers, we did not include these results or discussion in the revised manuscript. The comparisons between in-hostanalyses with cross-sectional analyses, as related to age, sex and disease severity, is an interesting knowledge gap. We hope that additional studies permit these comparisons.

3) Lines 366-:"We used regression analysis to assess the respiratory shedding of SARS-CoV-2 and compare age, sex or severity groups. " Please specify the type of regression analysis (I guess this was "normal" linear regression assuming normally distributed erros). How are measurements with undetectable virus loads included in the analysis? Wouldn't these measurements call for the use of censored regression models?

The regression was performed via normal linear regression and viral loads were taken at the detection limit of the assay. This analysis and consideration has been used to assess the trends of SARS-CoV-2 shedding between 1-10 DFSO throughout the literature, including the following studies:

Lucas, C., Wong, P., Klein, J. et al., Longitudinal analyses reveal immunological misfiring in severe COVID-19. Nature 584, 463–469 (2020). [Figure 3A]

Hurst, J.H., Heston, S.M., Chambers, H.N. et al., Severe Acute Respiratory Syndrome Coronavirus 2 Infections Among Children in the Biospecimens from Respiratory Virus-Exposed Kids (BRAVE Kids) Study, Clin. Infect. Dis., ciaa1693 (2020). [Figure 3B]

We have updated the manuscript to specify the type of regression analysis and included citations:

“To assess the respiratory shedding of SARS-CoV-2 and compare age, sex or severity groups, we analyzed the data using normal linear regression with interaction (Hurst et al., 2020; Lucas et al., 2020).” (page 19, line 523-524)

This was also specified in the body text to help with reader interpretation of our methods:

“To interpret the complex interplay between SARS-CoV-2 shedding dynamics and age, sex and COVID-19 severity, we stratified our systematic dataset into age, sex and severity groups and then conducted a series of linear regression analyses with interaction (Methods).” (page 6, line 136-138)

We also specified this in figure and table captions (Figure 4, Figure 5, Appendix Table 3 and Appendix Table 4).

4) The approach of how the authors estimated the AUC-ROC for the prediction of disease severity is not fully clear. Why did the authors need to first fit a distribution to the viral load values? The ROC curve and its AUC value can be computed directly from the observations, without this intermediate step.

We did so to determine the estimated parameters of these parametric distributions and the 95% CIs on the distribution. This was done to facilitate the interpretation of the distributions of rVL and their comparison between severity groups.

Moreover, we have made the code generated in this study available, including that which performed the AUC assessments. The Data availability statement includes:

“The code generated during this study is available at GitHub (https://github.com/paulzchen/sars2-shedding).”

5) Line 381-382: You write "Regression models were extrapolated (to 0 log10 copies/ml, rather than an assay detection limit) to estimate the duration of shedding." How large is the effect of extrapolating to 0 instead of to the detection limit? Some discussion on this is warranted.

We used our regression models to calculate this effect and added a discussion to the Methods section:

“Regression models were extrapolated to 0 log10 copies/ml to estimate the total duration of shedding. Some clinical studies report shedding duration based on assay negativity, when the viral RNA concentration in the specimen reaches the detection limit of the assay (often between 1-4 log10 copies/ml), and these cases may continue to shed viral RNA. To show the relationship between the two approaches, we used our regression model for URT shedding and estimated the shedding duration to a specimen concentration of 3 log10 copies/ml when sampling was conducted with nasopharyngeal swabs (approximately equivalent to a rVL of 2.1 log10 copies/ml). Then, the estimated mean duration of URT shedding for severe cases was 20.8 (95% CI: 14.5-27.0) DFSO, while it was 20.3 (95% CI: 16.8-23.7) DFSO for nonsevere cases. These values are in line with those reported by studies considering the assay detection limit (Cevik et al., 2020), supporting our regression models, and can be compared with those reported in the body text.” (page 20, line 554-565)

6) Line 390: Why was the Weibull distribution chosen? Some reasoning for this could be added to the paper (in methods, discussion or supplement)

We have included this reasoning in the methods section:

“Previously, our analyses found that SARS-CoV-2 rVLs best conform to Weibull distributions (P. Z. Chen et al., 2021).” (page 20, line 571-572)

7) Line 399-400: "The fitted Weibull distributions were used to estimate the accuracy when using URT or LRT rVLs of SARS-CoV-2 as a prognostic indicator for SARS-CoV-2 infection." Do you mean severity of infection here?

We mean COVID-19 severity. We have updated this sentence:

“The fitted Weibull distributions were used to estimate the accuracy when using URT or LRT rVLs of SARS-CoV-2 as a prognostic indicator for COVID-19 severity.” (page 21, line 586-587)

8) Table 1: There are relatively few lower respiratory samples included in the analysis, compared to the number of upper respiratory samples. What is typically the reason that a LRT sample is taken from a patient? And following, is there any possible association of taking an LRT sample with patient characteristics included or excluded in this study that could influence the results? Then some discussion on this would be warranted.

In our review, 6 out of 26 studies reported LRT samples. Four of these 6 studies used consecutive enrollment, but not all patients enrolled in these studies had a LRT sample. As the reviewers note in a comment below, the standard sample to detect SARS-CoV-2 is a nasopharyngeal swab. In the 6 studies, LRT samples were taken to detect SARS-CoV-2 infection as an alternative specimen type and/or to characterize the clinical features of COVID-19. Therefore, LRT samples may have been more often taken in complex patients at centres of excellence in respiratory care. To help account for potential selection biases, LRT sample data were stratified by disease severity, and risk of bias was evaluated and reported for each study. To further discuss this, we have included the following limitation:

“the review found that expectorated sputum was the predominant LRT specimen used for SARS-CoV-2 quantitation, and our analyses on LRT kinetics may not generalize to cases without sputum production.” (page 14, line 405-407)

9) Figure 4: Could it be made clearer in the figure which panels concern URT and which LRT? Now the reader has to carefully read the caption in order to deduce this.

To improve the ease of reader interpretation of Figure 4, we have added a line to separate the upper panels (which concern URT) from the lower panels (which concern LRT), added labels of “URT” and “LRT” in bold text in the figure, and amended the figure caption to better describe which panels correspond to URT and LRT analyses. Please see the revised version of Figure 4.

Reviewer #2 (Recommendations for the authors):The study appears robust and comprehensive, and relevant quality checks for systematic review have been applied. The results are valuable and contribute to the scientific knowledge in this field.Interesting findings include:– Adult patients with severe disease had on average a somewhat higher upper respiratory tract viral load at 1 day from symptom onset than patients with non-severe disease. After this stratification for severity, respiratory viral loads did not differ significantly for age and sex. Rates of viral clearing were similar. Children and adults with non-severe disease had similar upper respiratory tract viral loads and viral clearance rates.– High and persistent lower respiratory tract shedding of SARS-CoV-2 was associated with severe but not non-severe illness. The difference in lower respiratory viral load for severe and non-severe cases was more pronounced than for upper respiratory tract viral loads. In contrast to the upper respiratory tract, viral clearance from the lower respiratory tract was more rapid in non-severe than in severe cases. Again, age and sex did not differ significantly after stratification for severity.– The authors then aimed to assess whether the observed difference in shedding in the first days after start of symptoms could be used to predict which people would develop more severe COVID-19. Typically, deterioration into severe disease only happens around 10 days from symptom onset. The authors conclude that upper respiratory tract viral shedding is so heterogeneous that its predictive capacity of disease severity is inaccurate. In contrast, lower respiratory tract shedding does have a predictive accuracy of up to 81% for disease severity.Potential impact: Lower respiratory tract viral load could thus potentially be used as an early warning for developing severe COVID-19. However, lower respiratory tract samples are not routinely taken, the standard nasopharyngeal swab is an upper respiratory sample. Some discussion on the practical applicability of this suggestion could enhance the paper's impact.

We have included additional discussion on the applicability of this:

“Thus, LRT shedding may predict COVID-19 severity, serving as a prognostic factor. As emerging evidence suggests that timing influences the efficacy of anti-SARS-CoV-2 therapies (O'Brien et al., 2021; D. M. Weinreich et al., 2021), early clinical decision making is crucial. […] Furthermore, our data suggest that sex and age may not significantly influence prognostic thresholds but that the time course of disease may. Prognostication should account for the dynamics of shedding, and both the rVL and DFSO of a sputum specimen should be considered.”